# Numerical Analysis of Soil Water Dynamics during Spinach Cultivation in a Soil Column with an Artificial Capillary Barrier under Different Irrigation Managements

**Davy Sao** [1] , **Hirotaka Saito** [1,*] , **Tasuku Kato** [1] **and Jirka Šimůnek** [2]

1    Department of Agricultural and Environmental Engineering, United Graduate School of Agricultural Science, Tokyo University of Agriculture and Technology, 3-8-5 Saiwai-cho, Fuchu-shi, Tokyo 183-8538, Japan; davy_sao@hotmail.com (D.S.); taskkato@cc.tuat.ac.jp (T.K.)

2    Department of Environmental Science, University of California, Riverside, CA 92508, USA; jsimunek@ucr.edu

\*    Correspondence: hiros@cc.tuat.ac.jp

**Abstract:** Artificial capillary barriers (CBs) are used to improve root zone conditions as they can keep water and nutrients in the root zone by limiting downward percolation. Numerical analysis is one of the promising tools for evaluating CB systems' performance during the cultivation of leafy vegetables. This study aims to investigate the effects of the CB system on soil water dynamics during spinach cultivation in a soil column under different irrigation scenarios using HYDRUS (2D/3D) by comparing uniform (UNI), line-source (LSI), and plant-targeted (PTI) irrigations combined with alternative irrigation schedules. Simulation results of volumetric soil water contents were generally corresponding to measured data. Simulation results with various hypothetical irrigation scenarios exhibited that the CB was an effective system to diminish percolation losses and improve the root zone's soil water storage capacity. On the other hand, evaporation loss can be increased as more water is maintained near the surface. While this loss can be significantly minimized by reducing the water application area, the irrigation amount must be carefully defined because applying water in a smaller area may accelerate downward water movement so that the water content at the CB interface can reach close to saturation. In addition to the malfunction of the CB layer, it may also cause a reduction of plant root water uptake (RWU) because the root zone is too wet. Among evaluated irrigation scenarios, irrigating every two days with PTI was the best scenario for the spinach as water use efficiency was greatly improved.

**Keywords:** HYDRUS (2D/3D); capillary barrier; irrigation scenarios; soil water dynamics; root water uptake; water use efficiency





## 1. Introduction

Global demand for agricultural production, which is projected to rise as population and incomes are increasing [1], will directly affect agricultural water usage. Since water resources have been increasingly under pressure due to competition with other sectors, climate variability, socio-economic changes, overpopulation, together with unsustainable agricultural practices will put greater pressure on these limited resources [1]. To maintain or increase agricultural production, especially in water-scarce regions where water shortage is a major concern, it is important to develop reliable, simple, and cost-effective technologies to improve crop water use efficiency. An artificial capillary barrier (CB), a natural barrier for soil water flow under unsaturated conditions, is one such option as it can improve crop water use efficiency when properly designed and installed [2,3].

A CB happens at the joint boundary between two soil layers with different hydraulic properties, such as an interface of a fine-textured soil overlaid a coarse-textured soil. Because of differences in pore size distributions, capillary forces in the finer layer are greater than those in the coarser layer. When a finer particle soil overlays a coarser particle soil, this

phenomenon causes downward movement of infiltrating water to stop temporally at the interface until the matric potential at the interface reaches a value at which the coarser layer becomes more conductive. The effects of capillary barriers on water flow in soil profiles have been extensively studied. Stormont and Morris [4] and Morris and Stormont [5] used CB systems in landfill covers to redirect and prevent water from infiltrating into a sensitive sub-layer. Ityel et al. [2] installed a CB system to regulate infiltration rate and increase the root zone water retention capacity. The system was also used by Rooney et al. [3] to avoid the capillary rise of saline water from contaminated lower soil layer into the topsoil by placing a CB layer in between.

Numerical simulations have been recognized as a cost-effective tool and are increasingly used to examine soil water dynamics for optimizing irrigation management [6,7]. Among numerical models, the HYDRUS models are one of the most reliable and complete software packages that have been widely used and thus validated over the years [8]. The HYDRUS models have been successfully employed to study the effects of CBs on unsaturated soil water flow processes. For instance, Ityel et al. [2] confirmed HYDRUS (2D/3D) accuracy in assessing soil water contents in the CB soil system and used the model to identify the impacts of the CB on improving root zone conditions for the cultivation of horticultural crops. Kämpf et al. [9] fruitfully used HYDRUS-2D to conceptualize and parameterize flow processes in the CB system. Fala et al. [10] employed HYDRUS-2D to numerically simulate flow and the influence of CB in unsaturated waste rock piles. Henry [11] applied the HYDRUS-2D model to evaluate the degradation of CB performance due to the contaminant-induced surface tension reduction. Wongkaew et al. [12] confirmed the performance of HYDRUS-1D in predicting soil water contents and evaluated the impacts of alternative irrigation schedules under uniform irrigation on root zone moisture conditions in a CB soil column. Most recently, Al-Mayahi et al. [13] studied the influences of soil substrates made of Smart Capillary Barrier Wick, containing silt loam blocks enclosed by sand-sheaths and irrigated with a sand wick cylinder.

HYDRUS (2D/3D) has also been a popular model for assessing various irrigation systems [8]. The 3D model has been extensively used in modeling line-source and point-source irrigations in both homogeneous and heterogeneous soils. For instance, Elmaloglou et al. [14] successfully used the 2D model to estimate soil water fluxes under surface drip irrigation from equidistant line sources. Shakhali et al. [15] used HYDRUS-2D to simulate changes in volumetric soil water contents (VWCs) underline sources with different irrigation water salinities and reported an excellent performance of the model compared with field observations. Arbat et al. [16] also confirmed the reliability of the model to assess soil water dynamics, such as deep drainage and plant water uptake for various dripline depths, soil types, and irrigation schedules. Honari et al. [17] also reported high accuracy of the 3D model in modeling soil water dynamics under subsurface drip irrigation. Dabach et al. [18] successfully used HYDRUS (2D/3D) to evaluate the optimal placement of tensiometer for high-frequency subsurface drip irrigation in heterogeneous soils, while Mubarak et al. [19] demonstrated the accuracy of the 2D model in the study on the influences of high-frequency pulsing of drip irrigation in heterogeneous soils. However, the CB system's effects under different irrigation management (e.g., line-source and plant-targeted irrigations) have not yet been investigated.

This study investigates the effects of different irrigation practices, including uniform, line-source, and plant-targeted irrigations, on soil water dynamics during Japanese mustard spinach (Brassica rapa var. perviridis) cultivation in a soil profile with a CB layer installed. Alternative irrigation schedules were also evaluated to determine whether water use efficiency could be further improved while maintaining root water uptake by the plant. In a previous study, HYDRUS-1D successfully predicted soil water dynamics in the CB soil profile under spinach cultivation [12]. However, the one-dimensional model cannot simulate spatially non-uniform irrigations, such as line-source irrigation and plant-targeted irrigation. To achieve these objectives, numerical simulations for different irrigation types and alternative irrigation schedules were performed in a 3D domain. The applicability

of the HYDRUS (2D/3D) model was first evaluated by simulating changes in volumetric soil water contents (VWCs) and water fluxes in the experimental CB system under the spinach cultivation before various additional hypothetical alternative irrigation scenarios were simulated.

## 2. Materials and Methods

### 2.1. Cultivation Experiment

The cultivation experiment was carried out from 17 December 2013, to 14 January 2014, in a controlled environment with a temperature of 25 °C during daytime and 15 °C during nighttime [20]. Japanese spinach, one of the common leafy vegetables widely grown in Japan, was cultivated in four containers with a dimension of 24 cm wide × 36 cm long × 23 cm deep. One container, used as a reference, was filled with Tottori Dune sand, having an average particle size of 0.35 mm with 85% of particles between 0.106 to 0.425 mm, to a total depth of 16 cm (Figure 1b). Three other containers contained a CB system, as shown in Figure 1a. A CB layer, consisting of a 2-cm layer of coarse sand with the particle size between 0.85 and 2.00 mm and a 4-cm layer of gravel with the particle size between 4.75 and 9.00 mm, was placed between the top and bottom Tottori Dune sand layers making a thickness of 6 cm of the Dune sand at the top for plant root development and 4 cm of the Dune sand remaining at the bottom. The top 6-cm Tottori Dune sand layer in both reference and CB containers were mixed with fertilizers before seeding. A week after seeding, liquid fertilizers were additionally applied twice a week together with irrigation water.

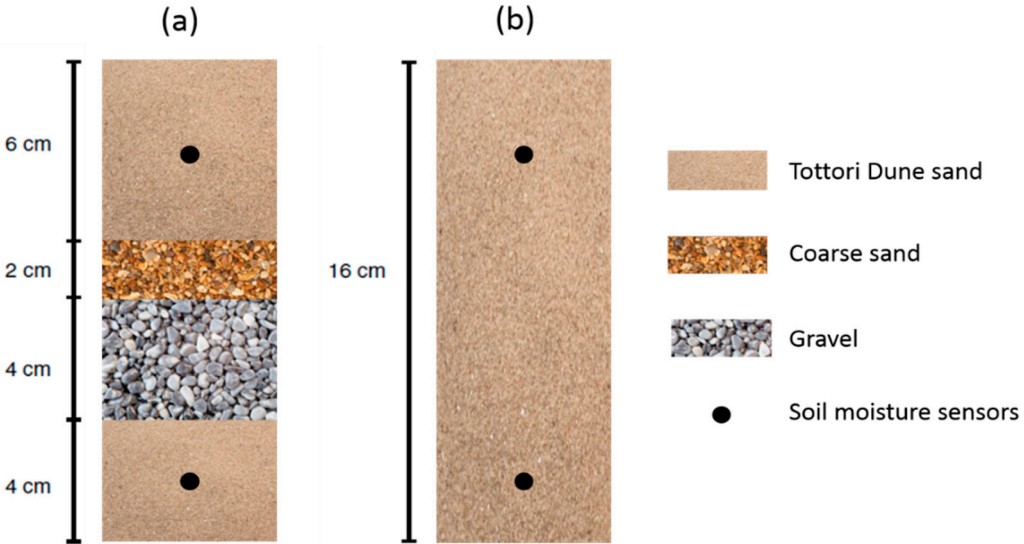

**Figure 1.** Soil profiles in the cultivation experiment of Miyake et al. [20] with (**a**) and without (**b**) a capillary barrier layer.

Irrigation at different rates was applied daily to each container uniformly at the surface using a sprinkling pot. The amount of water applied to the three containers with CB was 0.15, 0.30, and 0.45 cm per day, hereinafter called CB15, CB30, and CB45, respectively. The irrigation rate at the reference container (referred to as Non-CB30) was 0.30 cm per day. The water depth of 0.50 cm was applied on both the seeding day and the day after (the first day after seeding) to promote germination, while no irrigation was applied at all plots on the second day after seeding. The CB45 data were not analyzed further in this study since the irrigation amount in the CB45 was too high and caused the break in a CB, as reported by Miyake et al. [20] and also confirmed by the model simulation.

### 2.2. Numerical Simulations

The HYDRUS (2D/3D) model [8,21,22], a computer software package for simulating water, solute, and heat movement in variably saturated media, was used in this study

to evaluate spatial and temporal changes in volumetric soil water contents, soil water fluxes, and root water uptakes (RWU) under different irrigation designs and schedules in a three-dimensional (3D) domain. HYDRUS (2D/3D) numerically solves the Richards equation for saturated and unsaturated water flow in the 3D domain based on a finite element scheme. A sink term is included in the flow equation to consider plant root water uptake, which can be calculated using the Feddes RWU model [23]. Details about model simulation steps, as well as others required input data/parameters, are discussed below.

### 2.2.1. Soil Hydraulic Properties

In this study, similarly as in Wongkaew et al. [12], the single-porosity (SP) model of van Genuchten [24] was used to describe soil hydraulic properties of Totorri Dune sand, while the dual-porosity (DP) model of Durner [25] based on the van Genuchten [24] model was used for coarse sand and gravel. The SP model may not be appropriate to describe water flow in the dry range of relatively coarse materials, where water flows as non-capillary type flow after capillary water is drained. Therefore, the DP model, which can be used to mimic non-capillary type flow by suppressing the sharp decrease in the hydraulic conductivity in the dry range, was employed for coarse sand and gravel in this study.

To determine soil water retention curves of these materials, a modified suction table method [26] was used to measure water contents at predetermined matric potential heads ranging from 0 to −100 cm (Figure 2a). A pore size distribution model of Mualem [27] was used in both models to estimate unsaturated hydraulic conductivities (Figure 2b). The soil hydraulic properties, i.e., the soil water retention curve, $S_e(h)$, and the unsaturated hydraulic conductivity function, $K(h)$, for the SP model of van Genuchten [24] and the DP model of Durner [25] were described using Equations (1) and (2) and Equations (3) and (4), respectively.

$$S_e(h) = \frac{\theta - \theta_r}{\theta_s - \theta_r} = \frac{1}{\left(1 + |\alpha h|^n\right)^m}; \; h < 0 \tag{1}$$

$$K(h) = K_s S_e^l \left[1 - \left(1 - S_e^{1/m}\right)^m\right]^2 \tag{2}$$

$$S_e(h) = \frac{\theta - \theta_r}{\theta_s - \theta_r} = w_1 \frac{1}{\left(1 + |\alpha_1 h|^{n_1}\right)^{m_1}} + w_2 \frac{1}{\left(1 + |\alpha_2 h|^{n_2}\right)^{m_2}} \tag{3}$$

$$K(h) = K_s \frac{(w_1 S_{e_1} + w_2 S_{e_2})^l \left(w_1 \alpha_1 \left(1 - \left(1 - S_{e_1}^{1/m_1}\right)^{m_1}\right) + w_2 \alpha_2 \left(1 - \left(1 - S_{e_2}^{1/m_2}\right)^{m_2}\right)\right)^2}{(w_1 \alpha_1 + w_2 \alpha_2)^2} \tag{4}$$

where $S_e$ is the effective saturation (-), $\theta$ is the volumetric soil water content (VWC) ($cm^3/cm^3$), $\theta_r$ and $\theta_s$ are the residual and saturated volumetric soil water contents ($cm^3/cm^3$), respectively, $\alpha$ is the shape coefficient ($1/cm > 0$), $h$ is the soil water pressure head (cm), $n$ is a pore-size distribution index (-), $m$ is equal to $1 - 1/n$ ($0 < m < 1$) (-) (the parameters $\alpha$, $n$, and $m$ are fitting parameters affecting the shape of the retention curve), $l$ is a pore-connectivity parameter assumed to be 0.5 (-) [27], $K_s$ is the saturated hydraulic conductivity (cm/d), $w_1$ and $w_2$ are weighting factors for the two overlapping regions (-), and $\alpha_1$, $n_1$, $m_1$ ($= 1 - 1/n_1$), $\alpha_2$, $n_2$ and $m_2$ ($= 1 - 1/n_2$) are fitting parameters of the sub-curves.

The constant head method was used to measure saturated hydraulic conductivities ($K_s$) of dune sand and coarse sand. For gravel, $K_s$ could not be measured accurately using the same method because of the too high seepage rate. Therefore, it was calculated using the available empirical model of Hazen [28], following Wongkaew et al. [12]. Based on the Hazen equation, $K_s$ (cm/s) can be calculated using Equation (5).

$$K_s = C_H D_{10}^2 \tag{5}$$

where $C_H$ is an empirical coefficient $(1/(cm/s))$, which ranges from 1 to 1000 [29], and $D_{10}$ is the effective particle size (cm). Following Wongkaew et al. [12], $C_H$ was set to 1, and $D_{10}$ was a geometric mean of ten different $D_{10}$ values, ranging from 0.475 to 0.485 cm. The basic soil hydraulic parameters of the materials used in the HYDRUS (2D/3D) simulations are listed in Table 1.

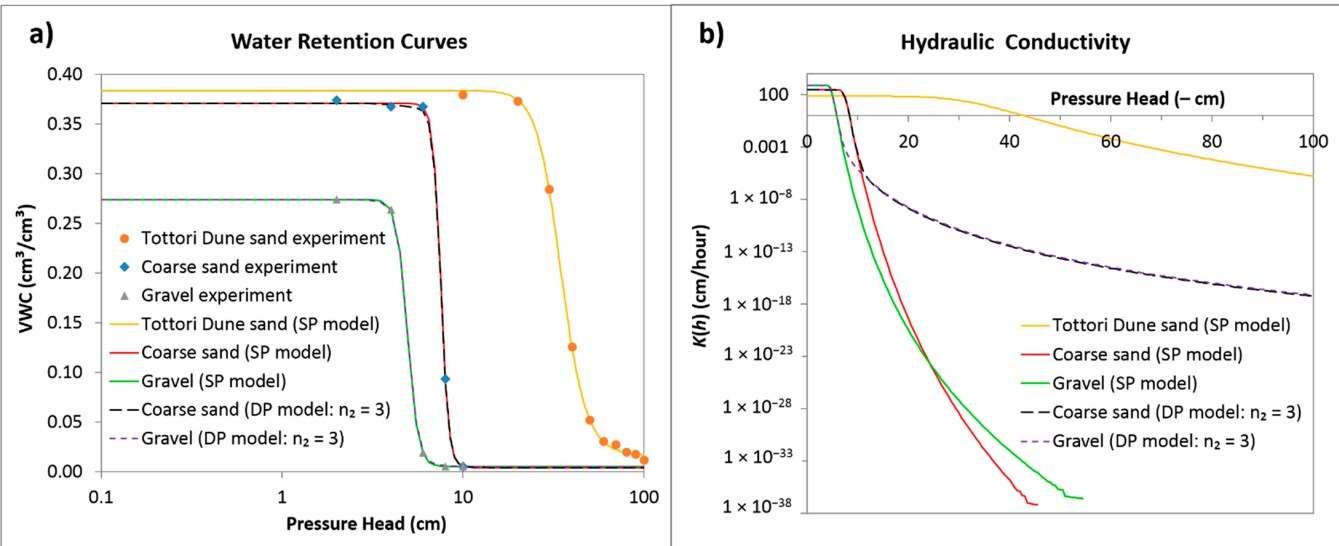

**Figure 2.** (**a**) Measured and fitted retention curves and (**b**) corresponding hydraulic conductivity functions of Tottori Dune sand, coarse sand, and gravel. The single (SP) and dual (DP) porosity models with $n_2 = 3$ were used [12].

**Table 1.** Soil hydraulic parameters for the single-porosity (SP) and dual-porosity (DP) models for Tottori Dune sand, coarse sand, and gravel [12].

| Materials | $\theta_r$ | $\theta_s$ | $\alpha$ (1/cm) | $n$ | $K_s$ (cm/s) | $w_1$ | $w_2$ | $\alpha_2$ (1/cm) | $n_2$ |
|---|---|---|---|---|---|---|---|---|---|
| Tottori Dune sand | 0.016 | 0.384 | 0.030 | 6.66 | 0.020 | - | - | - | - |
| Coarse sand | 0.004 | 0.371 | 0.133 | 20.8 | 0.081 | 0.99 | 0.01 | 0.25 | 3 |
| Gravel | 0.005 | 0.274 | 0.204 | 15.5 | 0.230 | 0.99 | 0.01 | 0.25 | 3 |

### 2.2.2. Root Water Uptake (RWU) Model

In HYDRUS (2D/3D), the form of the Richards equation was modified and adapted for the governing flow equation with the consideration of 2D/3D isothermal Darcian flow of water in a variably-saturated rigid porous medium by assuming that the air phase does not play a significant role in the liquid flow process. The equation is described as follows:

$$\frac{\partial \theta}{\partial t} = \frac{\partial}{\partial x_i}\left[ K(h)\left( K_{ij}^A \frac{\partial h}{\partial x_j} + K_{iz}^A \right) \right] - S \tag{6}$$

where $x_i$ ($i$ = 1, 2, and 3) are the spatial coordinates of the Cartesian coordinate system (cm), $t$ is time (day), $K_{ij}^A$ are components of a dimensionless anisotropy tensor $K_A$, and $S$ is a sink term (1/d). Anisotropy in soil hydraulic properties was not considered in this study.

The sink term, $S$, represents the water volume removed from the soil in a unit volume per unit time because of plant water uptake and is described by Feddes et al. [23] as:

$$S(h) = \alpha(h)S_p \tag{7}$$

where the water stress response function $\alpha(h)$ is a prescribed dimensionless function (-) of the soil water pressure head $h$ ($0 \leq \alpha \leq 1$), and $S_p$ is the potential water uptake rate (1/d). Notice that water uptake is assumed to be zero close to saturation (i.e., wetter than

"anaerobiosis point", $h_1$). For $h < h_4$ (the wilting point pressure head), the water uptake is also assumed to be zero. Water uptake is considered optimal between pressure head $h_2$ and $h_3$, whereas for pressure head between $h_3$ and $h_4$ (or $h_1$ and $h_2$), water uptake decreases (or increases) linearly with $h$. The variable $S_p$ is equal to the water uptake rate during periods of no water stress when $\alpha(h) = 1$.

In this study, because of the similarity in morphology with Japanese spinach, the Feddes parameters for lettuce were selected from the HYDRUS (2D/3D) database and adapted for the simulations. The initial threshold pressure heads of the lettuce's stress response function and their modified values used in this study are listed in Table 2. Root distribution parameters for the vertical and horizontal root distributions based on Vrugt's root distribution model [30] for Non-CB30, CB15, and CB30 scenarios shown in Table 3 were defined based on root data from Miyake et al. [20] and manually specified in the model. The HYDRUS (2D/3D) version used in this study did not allow to prescribe time-variable root growth, but only maximum values (e.g., the maximum rooting depth, the maximum rooting radius, etc.). The following Equation (8) describes the three-dimensional root distribution function of Vrugt et al. [30] implemented into HYDRUS.

$$b(x,y,z) \;=\; \left(1 - \frac{x}{X_m}\right)\left(1 - \frac{y}{Y_m}\right)\left(1 - \frac{z}{Z_m}\right)e^{-\left(\frac{P_x}{X_m}|x^* - x| + \frac{P_y}{Y_m}|y^* - y| + \frac{P_z}{Z_m}|z^* - z|\right)} \tag{8}$$

where $X_m$, $Y_m$, and $Z_m$ are the maximum rooting lengths in the $x$-, $y$-, and $z$- directions (cm), respectively; x, y, and z are distances from the origin of the tree in the $x$-, $y$-, and $z$- directions (cm), respectively; $P_x$ (-), $P_y$ (-), $P_z$ (-), $x^*$ (cm), $y^*$ (cm), and $z^*$ (cm) are empirical parameters ($x^*$, $y^*$, and $z^*$ are the coordinates of maximum root uptake intensity); and $b(x,y,z)$ denotes the three-dimensional spatial distribution of the potential root water uptake (-).

**Table 2.** Threshold parameters of the Feddes model used in the HYDRUS simulations for spinach.

| Threshold Parameters of the Feddes Model | Initial Parameters for Lettuce (cm) | Modified Parameters (cm) |
|:---:|:---:|:---:|
| $h_1$ (P0+) | −10 | −10 |
| $h_2$ (P0pt) | −25 | −25 |
| $h_3$H (P2H) | −400 | −50 |
| $h_3$L (P2L) | −600 | −50 |
| $h_4$ (P3) | −1200 | −100 |

As displayed in the HYDRUS GUI.

### 2.2.3. Domain for Simulation

The numerical simulations were conducted in a 3D simulation domain 16 cm deep, 6 cm wide, and 6 cm long (Figure 3). The size of the simulation domain, which was much smaller than the container used in the experiments, was determined to match the size of the soil profile corresponding to growing one spinach plant. The Non-CB domain consisted of a homogeneous uniform 16-cm layer of dune sand. The CB domain contained a 6-cm thick CB layer (a 2-cm layer of coarse sand overlying a 4-cm layer of gravel) located at a depth of 6 cm following the setup of the cultivation experiment (Figures 1 and 3). The spinach roots were assigned around the center of the simulation domain. The root density distributions at the cross-section of the center of the 3D model are depicted in Figure 3a. To compare with the observed VWCs, simulated observation points at the corresponding depths of 3 cm and 14 cm were assigned at the center of the transport domain.

**Table 3.** Parameters for the Vrugt root distribution model [30] for different experimental scenarios (Non-CB30—A homogeneous reference column and daily irrigation of 0.30 cm, CB15—A column with a CB and daily irrigation of 0.15 cm, and CB30—A column with a CB and daily irrigation of 0.30 cm) based on root data from Miyake et al. [20].

| Root Distribution Parameters | | Non-CB30 | CB15 | CB30 |
|---|---|---|---|---|
| Vertical Distribution | Maximum rooting depth | 7.2 cm | 2.8 cm | 4.3 cm |
| | Depth of maximum intensity | 2 cm | 2 cm | 2 cm |
| | Parameter pz | 1 | 1 | 1 |
| Horizontal Distribution—X | Maximum rooting radius | 2 cm | 2 cm | 2 cm |
| | Radius of maximum intensity | 0 cm | 0 cm | 0 cm |
| | Parameter px | 1 | 1 | 1 |
| | Center coordinate | 0 | 0 | 0 |
| Horizontal Distribution—Y | Maximum rooting radius | 2 cm | 2 cm | 2 cm |
| | Radius of maximum intensity | 0 cm | 0 cm | 0 cm |
| | Parameter py | 0 | 0 | 0 |
| Graphical distribution of the root system | | | | |

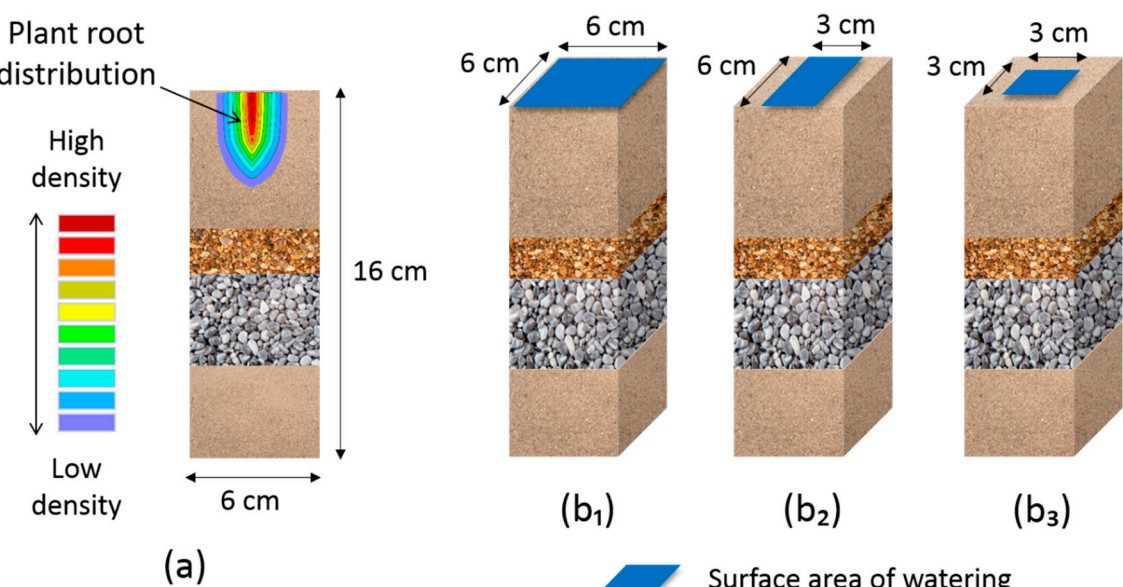

**Figure 3.** Schematic of root density distribution at the center cross-section of the simulation domain (**a**), and irrigation application types (uniform (**b₁**), line-source (**b₂**), and plant-targeted (**b₃**)).

### 2.2.4. Numerical Analysis of Cultivation Experiments

In the numerical analysis of the cultivation experiments, the observed irrigation fluxes were used as the surface boundary condition. Since the experiments were conducted in a controlled environment (a greenhouse), there was no rainfall. The actual evaporation

(E) and transpiration (T) rates (Figure 4) were adopted from Wongkaew et al. [12] by dividing the actual evapotranspiration (ET) data acquired from Miyake et al. [20] using the Surface Cover Fraction (SCF), which was measured once a week. Alternative irrigations were evaluated by assessing if these actual E and T rates were achieved when used as potential rates in the model simulations. The model also requires the minimum allowed pressure head value at the soil surface (hCritA), which defines the end of the first stage of evaporation. For numerical reasons, according to HYDRUS help, this value needs to be defined to make the corresponding water content at least 0.005 bigger than the residual water content, which is necessary particularly for coarse-textured soils, because of the steep slopes observed in their retention curves. A big value of hCritA may therefore be required for the stability of the numerical solution for the soils with coarse grain size. Also, when root water uptake is considered, a value of hCritA smaller (when negative) than P3 (i.e., the wilting point pressure head, $h_4$) should be taken. In this study, hCritA was set to $-250$ cm. For the boundary condition of the sidewalls and the bottom of the domain, a no-flux was assigned. No flow boundary at the sidewalls of the simulation domain was adapted, assuming that no lateral water flow interaction between plants occurred as uniform irrigation was applied in the experiments.

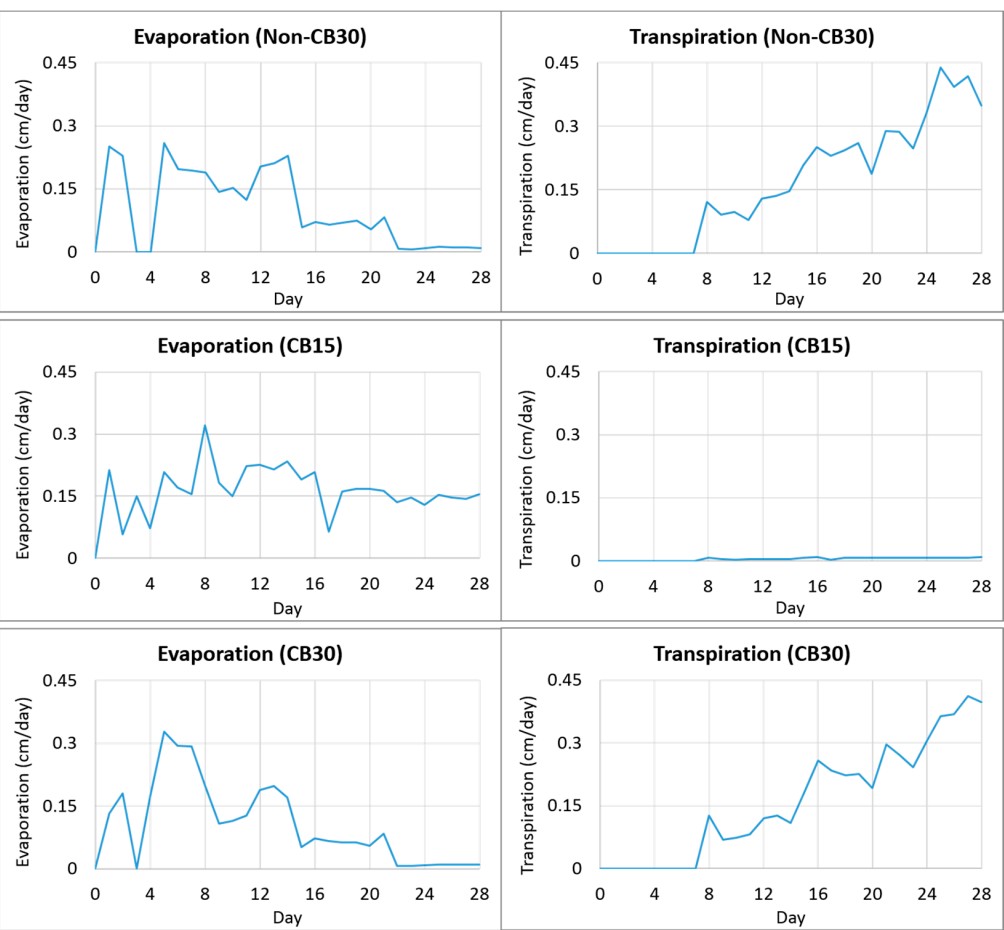

**Figure 4.** Evaporation and transpiration rates for Non-CB30 (**top**), CB15 (**middle**), and CB30 (**bottom**) adopted from Wongkaew et al. [12] by dividing the actual evapotranspiration rate obtained from Miyake et al. [20] for each simulation.

As an initial condition for the pressure head, a constant value of $-200$ cm was allocated to the entire domain. The model simulations were conducted from seeding until the final cultivation day, i.e., a total of 28 days. The initial time step was adjusted to a tiny value of $10^{-9}$ days to mitigate the effects of highly nonlinear soil hydraulic properties of

the coarse-textured soils. The minimum and maximum allowed time steps were set to $10^{-11}$ and $10^{-3}$ days, respectively.

The simulated time series of VWCs at two observation nodes (3 cm and 14 cm deep) were compared with the observed VWC values at corresponding locations to assess the model's accuracy. The squared Pearson correlation coefficient ($R^2$) and the root mean squared error ($RMSE$) were computed [31]. The $R^2$ value, varying between 0 (no correlation) and 1 (perfect correlation), defines how the observed and predicted values share the variance. The $RMSE$ determines how well the model performs by assessing the difference between observed and predicted values. The $RMSE$ is greater than or equal to 0, though the value of 0 (a perfect fit) is rarely attained in real life. Generally, the smaller the $RMSE$, the better the performance. The $R^2$ and $RMSE$ values were calculated according to Equations (9) and (10), respectively.

$$R^2 = \frac{\left[\sum_i \left(O_i - \overline{O}\right)\left(P_i - \overline{P}\right)\right]^2}{\sum_i \left(O_i - \overline{O}\right)^2 \sum_i \left(P_i - \overline{P}\right)^2} \tag{9}$$

$$RMSE = \left[\frac{1}{N} \sum_{i=1}^{N} (P_i - O_i)^2\right]^{0.5} \tag{10}$$

where $P_i$ is the $i^{th}$ predicted VWC value, $O_i$ is the $i^{th}$ observed VWC value, $\overline{P}$ is the mean predicted VWC value, $\overline{O}$ is the mean observed VWC value, and $N$ is the total number of observations. The $R^2$ is unitless, while the $RMSE$ units are the same as the units of $P$ and $O$.

### 2.3. Analysis of Alternative Hypothetical Irrigation Scenarios

The effects of the CB system on soil water dynamics and root water uptake during spinach cultivation were evaluated numerically under three different types of irrigation; uniform, line-source, and plant-targeted (referred to as UNI, LSI, and PTI, respectively). For each type of irrigation, three different irrigation schedules (referred to as Irri0, Irri1, and Irri2) were simulated to evaluate the influence of further reducing the amount of irrigation water on soil water dynamics. All simulations were conducted using the same simulation domain in 3D as done in the analysis of the cultivation experiments. In all simulations, a free bottom drainage boundary condition was used to allow water to drain freely at the bottom boundary and to quantify water loss to deep drainage.

Description of the three irrigation schedules mentioned above are as below:

- Irri0: Daily irrigation of 0.3 cm of water from Day 4 to the last day of the cultivation period;
- Irri1: Bi-daily irrigation of 0.3 cm of water during the first half of plant development stage from Day 4 to Day 14, and then daily irrigation until the last day of the cultivation period;
- Irri2: Bi-daily irrigation of 0.3 cm of water during the entire cultivation period from Day 4 to Day 28.

In all hypothetical irrigation scenarios, similarly as in the experiments, 0.5 cm of water was applied on Day 1 and Day 2, and no irrigation was applied on day 3.

In UNI, irrigation was applied uniformly on the whole surface (Figure 3b$_1$). In LSI, the same amount of water as in the UNI was applied around the plant on a surface area of 3 cm × 6 cm (Figure 3b$_2$). In PTI, the same amount of water was applied around the plant on a surface area of 3 cm × 3 cm (Figure 3b$_3$). The results of all these scenarios are discussed based on water balances and VWCs in the following sections. Note that UNI with Irri0 corresponds to either CB30 or Non-CB30 in the experiment, depending on whether a CB layer is installed or not. The only differences are the bottom boundary conditions; a free-drainage BC for hypothetical scenario simulations and a no-flux BC for experimental simulations. A summary of all simulation setups with their descriptions in both cultivation experiments and hypothetical irrigation scenarios is given in Table 4.

<div align="center">

**Table 4.** Summary of all simulation setups.

</div>

| Experiments/Scenarios | Bottom Boundary Condition | Irrigation | Simulation | Description |
|---|---|---|---|---|
| Cultivation Experiments | No-flux | Uniform | Non-CB30 | A homogeneous column (without CB) and daily irrigation of 0.30 cm |
| | | | CB15 | With CB and daily irrigation of 0.15 cm |
| | | | CB30 | With CB and daily irrigation of 0.30 cm |
| Hypothetical Irrigation Scenarios | Free drainage | Uniform, line-source, and plant-targeted | Non-CB_Irri0 | A homogeneous column (without CB) and daily irrigation of 0.30 cm |
| | | | CB_Irri0 | With CB and daily irrigation of 0.30 cm |
| | | | CB_Irri1 | With CB and bi-daily irrigation of 0.30 cm from Day 4 to Day 14, and then daily irrigation of 0.30 cm until the last day |
| | | | CB_Irri2 | With CB and bi-daily irrigation of 0.30 cm from Day 4 to Day 28 |

Since the Non-CB30 and CB30 cases were used as the benchmarks, the root distribution parameters and evapotranspiration rates obtained from the Non-CB30 experiment were used in the hypothetical scenario Non-CB_Irri0, while those from the CB30 experiment were used in hypothetical scenarios CB_Irri0, CB_Irri1, and CB_Irri2.

## 3. Results and Discussions

### 3.1. Simulation of Cultivation Experiments

3.1.1. Volumetric Soil Water Contents

Figure 5 shows simulated VWC profiles at the center of the 3D simulation domain for Non-CB30, CB15, and CB30, every 4 days between the start of the seeding period and the end of the experiment (Day 28). Observed VWCs at two depths of 3 cm and 14 cm are also plotted for comparison. The results indicated that, up to Day 4, the simulated VWCs in all experimental setups matched the observed data well at both observation depths. For the Non-CB30 setup, while the model predicted VWCs well at the lower observation point throughout the cultivation period, the model over-estimated VWCs at the upper observation point after Day 8. VWCs at the lower observation point were much higher than VWCs at the upper observation point, indicating that water accumulated at the bottom of the container in the Non-CB30 setup. In the CB15 and CB30 setups, the simulated VWCs at the lower observation points remained nearly constant at a relatively small value during the entire simulation period, while the observed VWCs slightly increased until Day 24 for CB15 and notably increased for CB30 until the last day. In the upper observation points, VWCs were generally overestimated for both CB15 and CB30.

Figure 6 shows scatter plots of simulated VWCs against observed VWCs at both observation points. For the Non-CB30 setup, simulations resulted in a small correlation ($R^2 = 0.044$) and a relatively high *RMSE* (=0.036) at the upper observation point. In contrast, simulations achieved a high correlation ($R^2 = 0.908$) and a small error (*RMSE* = 0.017) for the lower observation point for Non-CB30. For the CB15 system, the results showed a good correlation ($R^2 = 0.885$) with a reasonable error (*RMSE* = 0.027) at a depth of 3 cm and a very small error (*RMSE* = 0.013) at a depth of 14 cm with almost no correlation. The results for CB30 showed a large error (*RMSE* = 0.063) and a reasonable correlation ($R^2 = 0.607$) at a depth of 3 cm and relatively large errors (*RMSE* = 0.048) with almost no correlation at a depth of 14 cm, the depth which was beneath the CB layer.

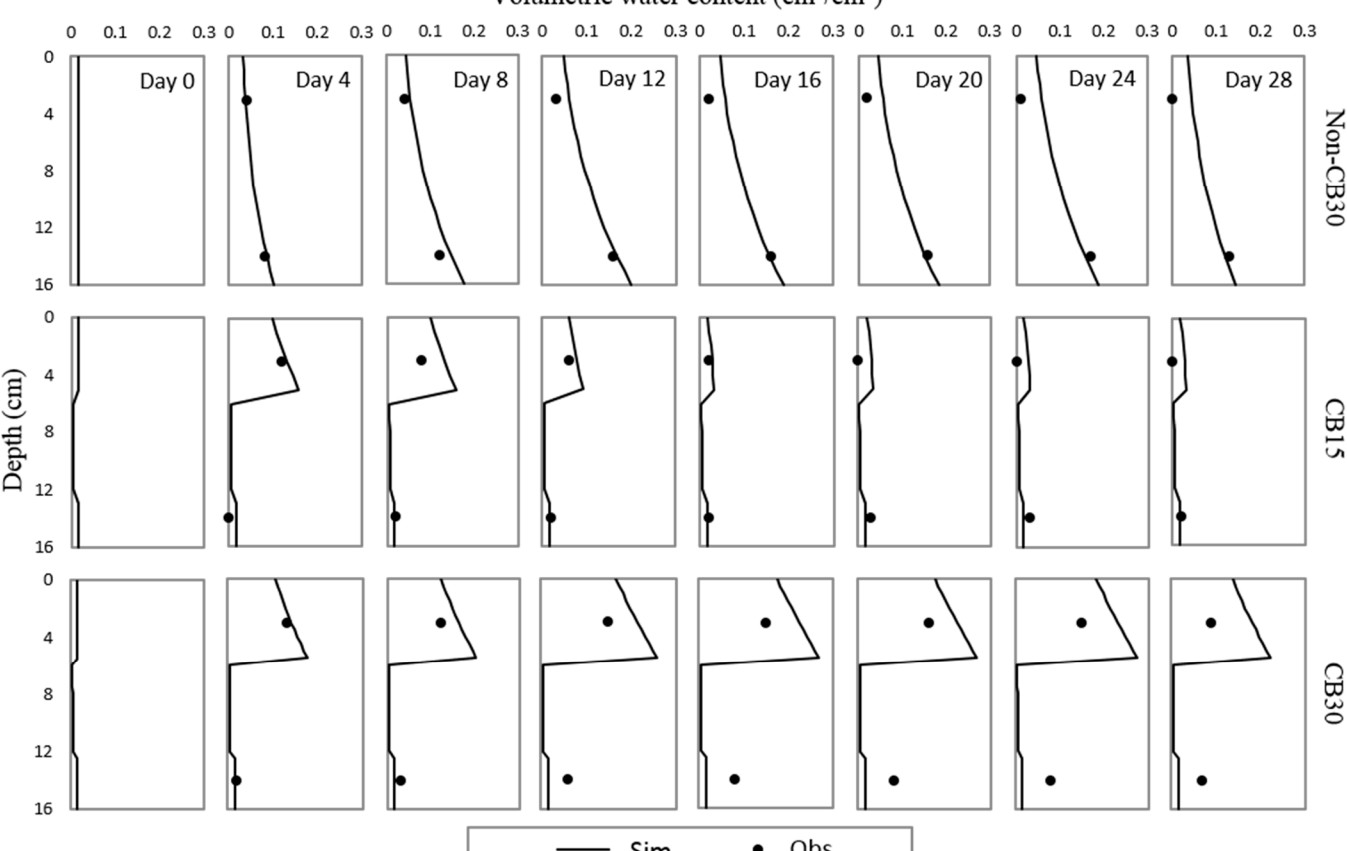

**Figure 5.** Simulated and observed volumetric soil water contents at different times (0, 4, 8, 12, 16, 20, 24, and 28 d; from left to right) and for different experimental setups (Non-CB30—A homogeneous reference column and daily irrigation of 0.30 cm (**top**), CB15—A column with a CB and daily irrigation of 0.15 cm (**middle**), and CB30—A column with a CB and daily irrigation of 0.30 cm (**bottom**)).

Although simulated results indicated that infiltrating water stopped at the CB interface (the VWCs were high in the top layer and almost zero in the bottom layer), the observed VWCs at the bottom of the containers increased slowly during the experiment, especially in CB30. This may have been due to water leaching along the containers' sidewalls into the bottom layer, which usually occurs in lysimeter studies [32–34]. Moreover, vapor flow towards the bottom containers may have happened because of the downward temperature gradient [20,35]. The soil surface temperature increased during the daytime while the sidewalls and the bottom boundary of all containers were insulated with styrene foams. In addition, although visual inspection could not detect obvious leakage of water through the CB layer, there could be liquid water flow through the CB layer in the experiments.

### 3.1.2. Cumulative Fluxes

Simulated cumulative fluxes (i.e., infiltration, evaporation, transpiration) for Non-CB30, CB15, and CB30 are summarized in Table 5. The Non-CB30 and CB30 setups produced similar water balance components: more than 50% of applied water was taken up by plant roots, almost 35% was lost via evaporation, and less than 15% was stored in the soil. For CB15, most of the irrigation water was lost due to evaporation (about 95%), while plant roots absorbed only less than 3%, and about 2% remained in the soil. These results imply that the CB system may enhance water losses through evaporation. When a small amount of irrigation water is applied, most of it can be returned to the atmosphere by evaporation.

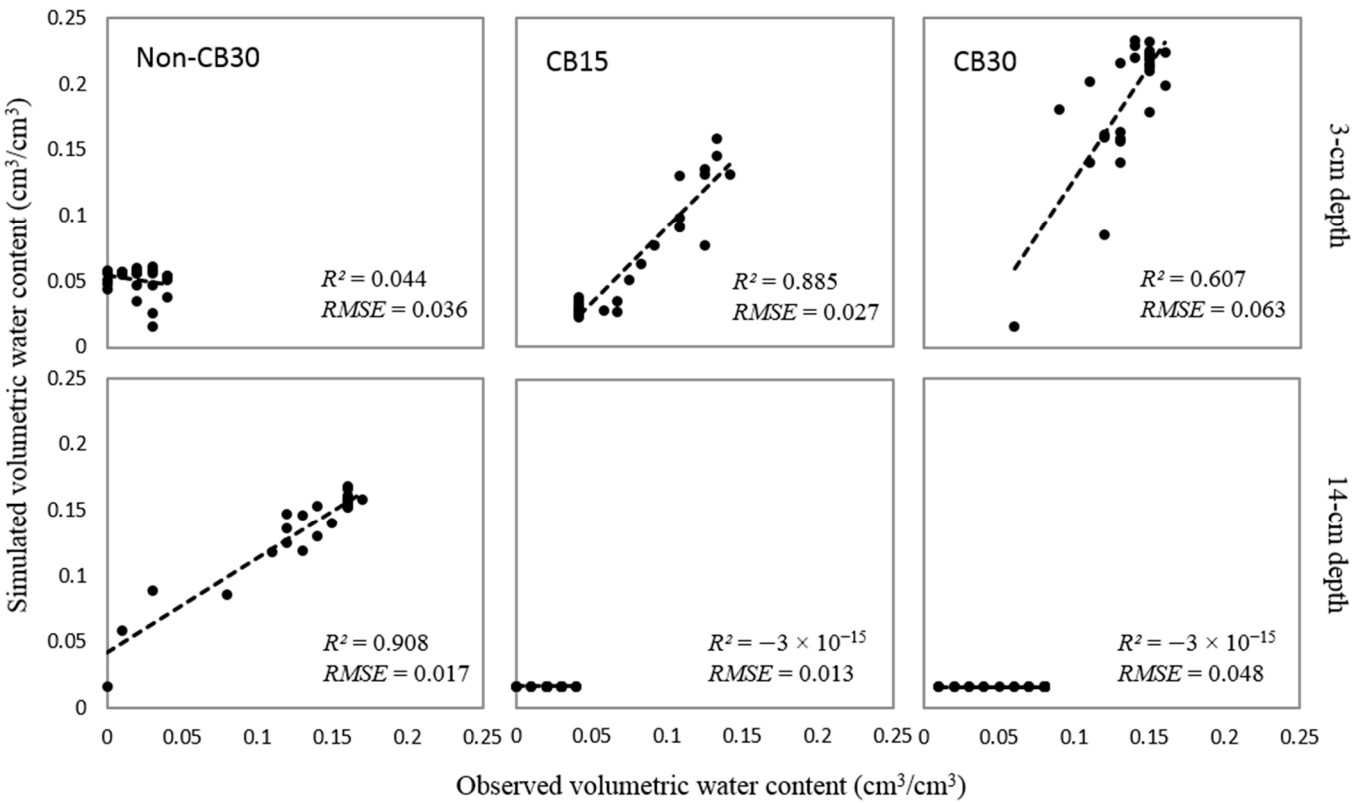

**Figure 6.** Scatter plots between simulated and observed volumetric soil water contents at depths of 3 (**top**) and 14 (**bottom**) cm for different experimental setups (Non-CB30—A homogeneous reference column and daily irrigation of 0.30 cm (**left**), CB15—A column with a CB and daily irrigation of 0.15 cm (**middle**), and CB30—A column with a CB and daily irrigation of 0.30 cm (**right**)).

**Table 5.** Simulated values of cumulative infiltration, evaporation, root water uptake (RWU), and soil water storage for different experimental scenarios (Non-CB30—A homogeneous reference column and daily irrigation of 0.30 cm, CB15—A column with a CB and daily irrigation of 0.15 cm, and CB30—A column with a CB and daily irrigation of 0.30 cm).

| Cumulative Fluxes | Non-CB30 (cm) | CB15 (cm) | CB30 (cm) |
|---|---|---|---|
| Infiltration (Irrigation) | 8.50 | 4.75 | 8.50 |
| Evaporation | 2.78 | 4.53 | 2.97 |
| Root water uptake (RWU) | 4.75 | 0.13 | 4.47 |
| Soil water storage | 0.97 | 0.10 | 1.06 |

To further illustrate the relationship between plant growth and simulated RWU, spinach heights (Figure 7a) and spinach's leaf area index (LAI) (Figure 7b) for all experimental setups (Non-CB30, CB15, and CB30) at the 2nd, 3rd, and 4th week of the plant growth obtained from Miyake et al. [20] are plotted against corresponding simulated cumulative RWU at corresponding days. The graphs show a good agreement between simulated root water uptake and the corresponding spinach growth, having high positive correlations between these two pairs of the variables, with an $R^2$ value of 0.93 for simulated RWU vs. spinach height and an $R^2$ value of 0.99 for simulated RWU vs. LAI. For instance, simulated RWU in Non-CB30 was similar to that in CB30, while the actual spinach growth conditions in the two experiments were nearly the same. A more obvious example is that simulated RWU in CB15 was only about 3% of that in CB30 or Non-CB30 (Table 5), while leaf area index (LAI) of the spinach in CB15 was no more than 3% of those in CB30 or Non-CB30 (Figure 7b). These results indicated that the evaluation of spinach growth based on the simulated RWU is dependable. Moreover, it was also observed that when the spinach

developed more leaves in CB30 and Non-CB30 and the surface cover fraction correspondingly increased, simulated evaporation rates in those two experimental simulations were low, meaning that water loss is mainly due to transpiration. On the other hand, when the spinach grew extremely small in CB15, simulated evaporation in that treatment was quite large.

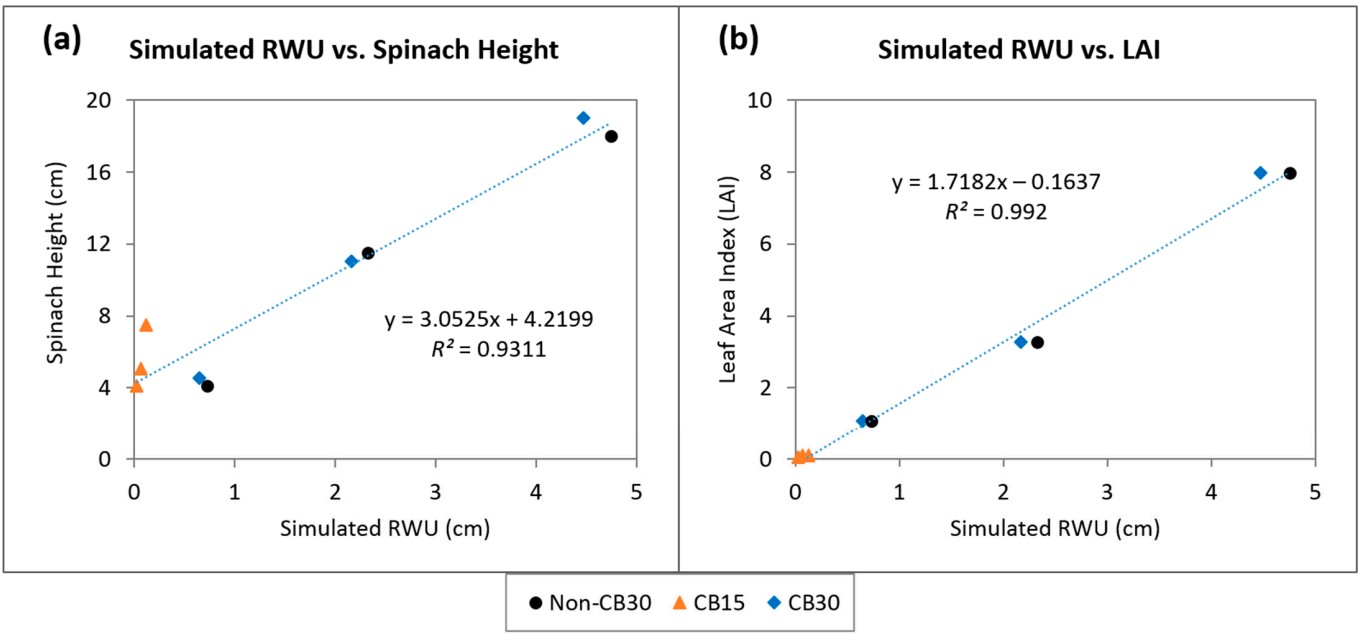

**Figure 7.** Scatter plot of cumulative RWU simulated by the model versus (**a**) spinach heights and (**b**) leaf area index obtained from Miyake et al. [20] at days 14, 21, and 28 for all experimental setups (Non-CB30, CB15, and CB30).

As simulated cumulative RWU reflects the spinach growth well, simulated RWU is used in the remainder of the manuscript to discuss the performance of various irrigation scenarios.

### 3.2. Simulation Results for Hypothetical Scenarios

3.2.1. Water Balances for Different Irrigation Scenarios

Figure 8 shows cumulative water fluxes for different irrigation scenarios simulated by the HYDRUS (2D/3D) model in a 3D domain. Simulated water balance components were found to vary depending on irrigation types (whether uniform, line-source, or plant-targeted), especially in the soil profiles with a CB. For Non-CB with Irri0 (Figure 8a), the three different irrigation types, UNI, LSI, and PTI, did not influence much simulated cumulative fluxes and produced almost no change in soil water storage (SWS). While cumulative evaporation was reduced from 1.3 cm for UNI to 0.8 cm for LSI, in which irrigation water was applied on a smaller part of the surface, both cumulative root water uptake (RWU) and cumulative bottom drainage (BD) slightly increased. When the same amount of water was directly applied around the plant in PTI, cumulative evaporation was further reduced to 0.5 cm with slight increases in RWU and BD.

On the other hand, simulated cumulative fluxes in soil profiles with a CB were strongly affected not only by irrigation types but also by irrigation schedules (Figure 8b–d). For instance, for CB_Irri0 with UNI, in which daily irrigation of 0.3 cm was applied uniformly, cumulative evaporation, cumulative RWU, and cumulative BD were 3.0 cm, 4.5 cm, and 0 cm, respectively, and SWS increased by 1.1 cm (Figure 8b). For LSI, while cumulative evaporation decreased to 1.5 cm and cumulative RWU dropped to 1.2 cm, cumulative BD and SWS increased considerably to 3.5 cm and 2.3 cm, respectively. The water balance components varied further for PTI, except for SWS. Cumulative evaporation for PTI was reduced by half compared to that for LSI. Additionally, while cumulative RWU decreased

to 1.0 cm, cumulative BD increased to 4.4 cm for PTI. Although irrigating 0.3 cm of water every day (CB_Irri0) was good for UNI as the water balance ratio was improved with higher cumulative RWU and smaller water losses compared to Non-CB, it might not be appropriate for LSI and PTI, as a huge amount of water was lost through bottom drainage with lower cumulative RWUs. The local applications of this amount of water could lead to an exceedance of the threshold pressure heads in the wet range of the Feddes transpiration function. This indicates that even with a CB system installed, applying irrigation on a small area with an excessive water amount could reduce root water uptake because the root zone is too wet and roots are under stress. Rising water content at the interface may produce breakage of the CB barrier and water losses through bottom drainage.

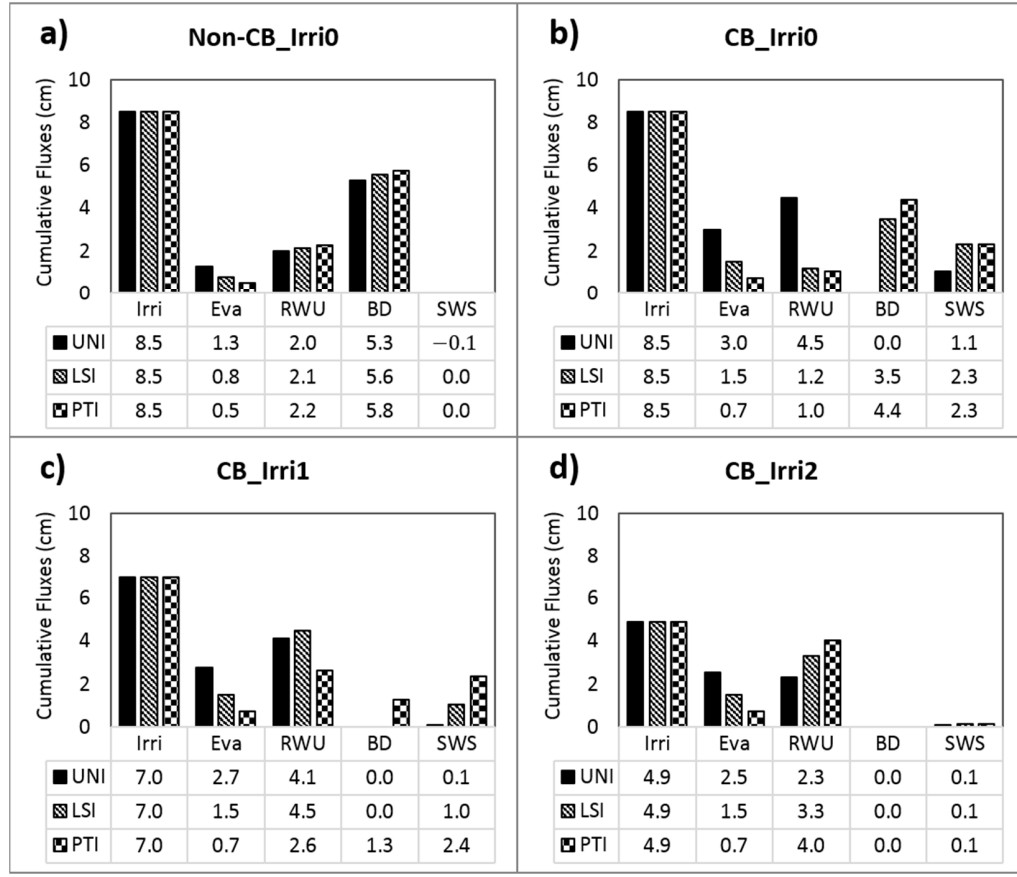

**Figure 8.** Cumulative water fluxes for uniform, line-source, and plant-targeted irrigation (UNI, LSI, and PTI, respectively) for different simulation setups: (**a**) Non-CB_Irri0, (**b**) CB_Irri0, (**c**) CB_Irri1, and (**d**) CB_Irri2. Free bottom drainage was used fot the bottom boundary condition. Note: Irri—irrigation; Eva—evaporation; RWU—root water uptake; BD—bottom drainage; and SWS—soil water storage.

For CB_Irri1 (Figure 8c), when irrigation water was reduced by applying water every two days during the first half of the plant growth stage, cumulative bottom drainage did not occur for UNI. Cumulative evaporation and cumulative RWU slightly decreased to 2.7 cm and 4.1 cm, respectively, while SWS was remarkably reduced to 0.1 cm compared to CB_Irri0 with UNI. Notable results were obtained when it came to LSI with CB_Irri1. Cumulative evaporation was further reduced to as low as 1.5 cm, while cumulative RWU and SWS increased to 4.5 cm and 1.0 cm, respectively, similar to the values obtained for UNI with CB_Irri0. However, for PTI, although cumulative evaporation was further reduced by half compared to cumulative evaporation for LSI and the SWS increased to 2.4 cm, cumulative RWU was considerably reduced to 2.6 cm, and cumulative BD increased

to 1.3 cm. Like CB_Irri0 with PTI, applying irrigation over a smaller area resulted in malfunctioning of a CB, as excess infiltrating water accumulated at the CB interface.

For CB_Irri2 with UNI (Figure 8d), in which irrigation water was further reduced by applying water every two days throughout the cultivation period, cumulative evaporation and cumulative RWU decreased to 2.5 cm and 2.3 cm, respectively. Cumulative BD and SWS were the same as for CB_Irri1 with the same irrigation type. For LSI with CB_Irri2, although there was no effect on cumulative BD and SWS, cumulative evaporation was reduced to 1.5 cm while cumulative RWU increased to as much as 3.3 cm. For PTI, on the other hand, even though the total amount of water applied was remarkably reduced compared to Irri0 and Irri1, cumulative RWU was attained at 4.0 cm, while cumulative evaporation decreased to 0.7 cm, and cumulative BD and SWS remained unchanged. This result indicates that because the total amount of water applied was much smaller, even though the water was applied locally around the plant, there was no excess water accumulated at the CB interface, and thus the CB functioned as designed.

Interestingly, although in CB_Irri2 with UNI, the same total amount and the same type of irrigation were applied as in CB15, cumulative RWU was much larger than that in CB15. This may be explained by the fact that applying 0.30 cm of water once in two days in CB_Irri2 resulted in the reduction of water losses by evaporation as water could infiltrate much deeper. When applying 0.15 cm of water every day, water loss by evaporation was enhanced.

This study demonstrated that in addition to drastically reducing water loss through evaporation, applying irrigation water around the plant, as done in LSI and PTI, could improve soil water storage capacity compared to UNI. However, when the water application area was reduced, the soil at the interface of the CB layer can be quickly saturated with excess infiltration water to cause malfunctioning of the CB system. Increased saturation in the soil may further inhibit RWU by reducing oxygen in the root zone. The plant growth can be thus depressed. The numerical analysis of hypothetical irrigation scenarios suggests that the total amount of water needs to be reduced if water is applied more locally when the CB system is installed to avoid its malfunctioning and plant growth depression.

3.2.2. Vertical VWCs by Different Scenarios

To further understand how irrigation scenarios affect soil water dynamics, temporal changes in simulated VWC profiles in the center of the simulation domain for all irrigation scenarios were plotted in Figure 9. VWCs were almost identical between different irrigation types for Non-CB_Irri0 as differences in irrigation application areas (either 6 cm × 6 cm for UNI, 3 cm × 6 cm for LSI, or 3 cm × 3 cm for PTI) produced only minimal differences in temporal changes in the VWC profiles (they were mostly overlapping as shown in this figure). The same trend was observed for cumulative fluxes (Figure 8a). On the contrary, for the soil profiles having a CB, the VWC profiles for the three irrigation types, UNI, LSI, and PTI, were visibly different from Day 4, regardless of the irrigation schedule (i.e., Irri0, Irri1, or Irri2). Figure 9 shows that the VWC at the top of the CB layer for PTI with CB_Irri0 reaches a level that would cause water to percolate through the CB system (to break the function of the CB system and to increase downward water percolation) after Day 8. The CB function was broken for LSI with CB_Irri0 from Day 12, while it was never broken for UNI with this irrigation schedule. Therefore, no bottom drainage for UNI was exhibited in Figure 8b.

The VWC profiles were all different for the three irrigation types with CB_Irri1. Only PTI with CB_Irri1 had water percolation through the CB layer after Day 20. Consequently, bottom drainage was only observed for PTI in this irrigation schedule, as presented in Figure 8c. The CB layer remained fully functioning regardless of the irrigation type with CB_Irri2. PTI provided the highest VWC in the root zone (i.e., the soil layer above the CB layer) until Day 24. At the end of the cultivation period (i.e., Day 28), the entire domain became completely dry for all irrigation types with Irri2. As a result, there was no

bottom drainage throughout the simulation period and a relatively small soil water storage regardless of the irrigation type (Figure 8d).

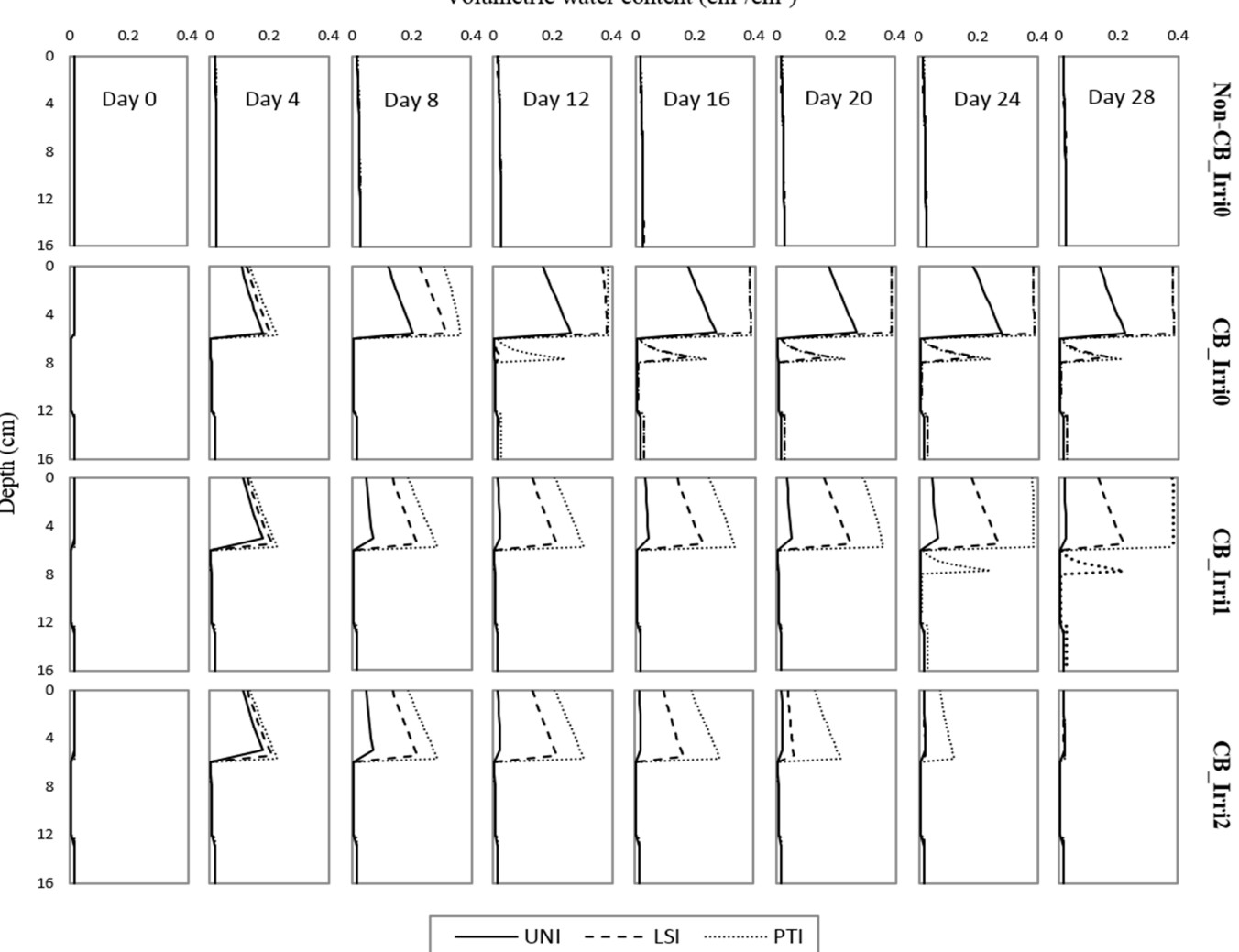

**Figure 9.** Profiles of volumetric soil water contents at the center of the domains obtained from the model for uniform, line-source, plant-targeted irrigation (UNI, LSI, and PTI, respectively) at different times (0, 4, 8, 12, 16, 20, 24 and 28 day) and for different simulation setups with free bottom drainage (Non-CB_Irri0, CB_Irri0, CB_Irri1, and CB_Irri2).

## 4. Conclusions

The HYDRUS (2D/3D) model was employed to evaluate soil water dynamics in soil profiles with a capillary barrier system installed during a cultivation experiment of Japanese spinach under different hypothetical irrigation scenarios. The performance of the HYDRUS (2D/3D) model was first evaluated by simulating the soil water dynamics in a 3D domain based on the cultivation experiment of Miyake et al. [20]. The results showed good correspondence between observed and simulated VWCs and confirmed the effectiveness of the CB layer to improve root zone conditions by suppressing downward water percolation.

Based on the numerical analysis of hypothetical irrigation scenarios, the following conclusions are drawn: (1) while a CB generally reduced water losses due to deep percolation and improved root zone water storage capacity during the cultivation, water losses due to evaporation could be enhanced as the water was retained in the topsoil layer by the CB system, (2) evaporation loss could be significantly reduced in the CB system by shrinking the irrigation application area, and (3) reduction of RWU and malfunctioning of the CB could occur when too much irrigation water was applied around the plant as the

excess water accumulated at the CB interface, thus causing breakage of the CB and stress to plant roots due to wetness.

The study suggests that irrigation types, amounts, and schedules need to be carefully determined to maximize water use efficiency during crop cultivation in a layered soil consisting of a CB layer. In this context, numerical simulations are helpful and can be used as a decision support system.

**Author Contributions:** Conceptualization, H.S. and D.S.; methodology, H.S. and D.S.; model simulations, D.S.; validation, D.S. and H.S; formal analysis, D.S. and H.S.; investigation, D.S. and H.S.; resources, H.S.; data curation, H.S.; writing—original draft preparation, D.S.; writing—review and editing, H.S., J.Š., T.K. and D.S.; visualization, D.S.; supervision, H.S., T.K. and J.Š.; project administration, H.S.; funding acquisition, H.S. All authors have read and agreed to the published version of the manuscript.

**Funding:** This study was partially supported by the Joint Research Program of Arid Land Research Center, Tottori University, and the JSPS KAKENHI Grant Numbers JP17H03885, JP18H02295, and JP20H03097.

**Institutional Review Board Statement:** Not applicable.

**Informed Consent Statement:** Not applicable.

**Data Availability Statement:** The data presented in this study are available on request from the corresponding author.

**Acknowledgments:** The authors would like to thank the anonymous reviewers for their comments to improve the initial manuscript.

**Conflicts of Interest:** The authors declare no conflict of interest. The funders had no role in the study's design, in the collection, analyses, or interpretation of data, in the writing of the manuscript, or in the decision to publish the results.

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
