# Peer review of "Numerical Analysis of Soil Water Dynamics during Spinach Cultivation in a Soil Column with an Artificial Capillary Barrier under Different Irrigation Managements"

_water, doi:10.3390/w13162176_

Round 1

Reviewer 1 Report

This paper gives a nice demonstration of the use of a soil water flow model to evaluate the effect of capillary barriers, irrigation scheduling and irrigation application methods (uniform, line source or locally around the plant) on the water balance components in the system. The simulation model was first validated against experimental data and subsequently used to simulate different irrigation scenarios. The simulations nicely demonstrate what the implication of non-linearities in soil water flow and root water uptake processes are on soil water balance components. For instance, applying less water may lead to more root water uptake, applying it uniformly rather than locally may in some cases lead to more and in other cases to less root water uptake. Introducing heterogeneities in the system like CB may even further complicate things. This paper nicely illustrates that using simulation models that describe these non-linear processes are required to make predictions on how different water balance components will change under different irrigation schedules and application methods.

The paper is overall very interesting and addressing a very relevant topic (water use efficiency in irrigation). Although the paper is well written, I think it still requires a few technical clarifications.

First, the simulation setup and how it matches with the experiment could be improved. I did for instance not understand why the dimensions of the experimental containers differed from those of the simulated ones.

Second, the parameterization of the plant development (root distributions, transpiration over time) and its role in the different water balance components need to be explained and documented better. For instance, there seems to be an inconsistency in the simulated transpiration of the experiment CB15 that does not seem to be consistent with the observed plant development. Giving more information on the simulated plant water stress (and the reasons for it: too wet or too dry soil conditions) is required to understand the impact of the irrigation regimes on the water balance.

Third, the interaction between plant development and soil water conditions seems to play an important role in the soil water balance components. However, plant development needs to be prescribed by the model as it does not simulate plant development. Therefore analyzing the effect of different irrigation scenarios would require considering how the plant growth reacts to it as well. In the current model setup, the impact of the soil conditions on the plant water stress is considered but this stress depends of course on the root distribution and the leaf area of the plant. I think the authors could address this point since it seems to me the reason why in the experiment with reduced irrigation, CB15, the model simulated almost no transpiration and hence severe stress whereas in the scenario with reduced irrigation, Irri2, more transpiration was simulated and hence less water stress. I suppose this was due to a different parameterization of the root distribution and leaf surface area in the experiment simulation than in the scenario simulation.  

Detailed comments:

Why was there a 4 cm thick sandy layer at the bottom?

Ln 119:  What is the depth, width and length of the containers?

Ln 124: ‘Fertilizers were mixed into the upper Tottori Dune sand layer before seeding.’ Was the same upper layer thickness considered in the reference as in the CB container?

Ln 136: It would have been interesting to analyze also the CB45 experiment since it would be interesting to know whether the simulation models also predicted a breaking of the CB.

Ln 178: The n values of these soils are relatively high. This makes that the water content and hydraulic conductivity drop sharply when the pressure head (its absolute value) reaches 1/alfa. In view of the parameterization of the root water uptake and evaporation that is discussed later, it would be illustrative to include the retention curves and conductivity curves of the soil.

Ln 203: Maybe it comes later but I am wondering how these root water uptake parameters were updated.

Table 3: The parameters of the Vrugt root distribution model are given here but as for the water retention function, also the function that describes the root distribution should be given. In  order to see the difference between the root distributions, I think also a plot of the root distributions should be given.

Ln 214: Why was a smaller domain simulated than the actual size of the container? Is it correct to assume no flow boundary conditions at the lateral boundaries?

Ln 229: You should give the potential transpiration and evaporation rates and how they evolved over the simulation period. I suppose transpiration increases over time as the plants develop. Was root growth considered and did the SCF change over time?

Ln 232: How is the surface cover fraction defined? Is that the area covered by the leaves?

Ln 251: R² is usually used as symbol for the coefficient of determination. But, Eq. 8 is not the equation that defines the coefficient of determination. In fact, Eq. 8 is the squared Pearson correlation coefficient. The coefficient of determination can take values smaller than 0. That is when the predictions by the model are a worse prediction than the mean of the observations. For least squares ordinary linear regression, it can be proven that the coefficient of determination corresponds with the squared correlation coefficient.

Ln 263: Alternative irrigation scenarios. Which root distributions were taken here and what plant development was assumed? For the CB30 and non-CB30, the plant development was quite similar (but not the root distribution). But, for CB15, the root development and shoot development was limited. In the Irri0 scenario, the same amount of water is applied as in the CB30 and non-CB30 scenarios. But less water is applied in the Irri1 and Irri2 scenarios. In Irri2, the same amount of water was applied as in CB15. Was the plant development and root distribution then adjusted in the Irri1 and Irri2 scenarios?

Discussion on figure 3 and 4. The authors discuss well the observed differences between the observations and the measurements. I just wanted to point out that there might also be uncertainties and errors in the measurements. Do the water content measurements close the water balance? The model does but what about the measurements? I don’t know how water contents were measured but there might also be a considerable uncertainty on water content measurements.

Ln 329: Water flow along the container side walls and vapor transport are put forward as reasons for the increase in the soil water content below the CB. But, can it be that there is water flow through the CB because the hydraulic properties of the CB are not exactly the same as the ones used in the simulations?

Ln 340: In CB15, the simulated transpiration was only 3% of the transpiration simulated for CB30 or non-CB30. If one would assume a constant water use efficiency of the spinach, then this would mean that the biomass of the spinach in the CB15 experiment was only 3% of that in the other two experiments. This would mean that the spinach in the CB15 would almost not have grown during the experiment. Therefore, I think that the transpiration rate is strongly underestimated (in relative terms) in the simulations of CB15. For the water balance calculations, this may probably not have caused a big impact since the lower transpiration could have been compensated by higher evaporation. But, in order to evaluate the plant growth in different irrigation scenarios in terms of the simulated transpiration, a good estimate of the transpiration is of course important. Using plant transpiration as a proxy for plant growth, this would mean that in CB15, the model would simulate almost no growth, which was apparently not the case.

It is interesting that in CB30 and non-CB30, simulated evaporation was lower than in CB15. But, was this because in these simulations, the higher transpiration led to lower surface water contents and hence an earlier reduction in evaporation rate? Or was the evaporation lower because the spinach developed more leaves and surface cover fraction by the leaves was higher in the CB30 and non-CB30 treatments? It is important to understand which are the drivers of the differences in the simulated water balance components: plant development and parameterization of evaporation or soil properties and water distributions in the soil

Figure 5: I propose to use different symbols for the different experiments. I suppose it will come out then that for the CB15 experiment, the model does not predict the relation between plant height and cumulative transpiration well. By lumping all the experiments in one plot, one might still get a good R² but this is then mainly because of the relation between the plant development and the transpiration in the experiments in which a lot of water was applied. But this does not mean that the transpiration was predicted well in the experiments in which less water applied. Or vice versa, that the growth could be predicted based on the simulated transpiration.  

Ln 377: Discussion on the water application method in Irri0 on transpiration and leaching. I think the authors should give some more explanation why the root water uptake is reduced so much when the water is applied as a line source or around the plant. According the authors, the local water application leads to a stronger local wetting and therefore more loss through the CB and therefore less root water uptake. But, it could also be that the local wetting leads to higher soil water pressure heads and therefore an exceedance of the threshold pressure heads in the wet range of the Feddes transpiration reduction function. So, the reduction in transpiration could also have been due to stress conditions which then lead to more water excess and breakthrough of the CB. It was not clear to me what the causal reason was of the reduction in transpiration: failure of the CB or root water uptake reduction due to too wet conditions.

Ln 406: Discussion on Irri2. In Irri2, the same amount of water was applied as in CB15. However, in CB15, transpiration was only 0.13 cm whereas in Irri2 with uniform water application, it was more than 2 cm. I do not know what the reasons for this difference are. Is that related to differences in root distribution and assumed leaf development in the two scenarios? If so, doesn’t this show then that in order to assess the effect of different irrigation regimes on the soil water balance, it is important to consider how the plant development reacts to the irrigation regimes.

Ln 471: In the model it is assumed that when only a fraction of the surface is wet and the other part is completely dry, that then the evaporation over the total area is proportional to the wet fraction of the surface. This assumption is however not correct because evaporation of wet patches in a dry surrounding is enhanced due to lateral heat flow in the soil and lateral advection in the air. So I would add here that the simulated reduction in evaporation might not be so large in reality due to these compensation effects which are not considered in the model.

Reviewer 2 Report

The paper presents results related to the capillary barrier simulations and lysimeter experimentation. It is well written, I can’t see any stronger deficiencies in the manuscript. 

I think it is worth publishing. 

Minor comments:

L66-67 Placing the link directly in the text of the manuscript looks somewhat odd to me. I’d suggest moving this website to the cited sources and user regular citation instead. 

L152-154 Could you add the rationale standing behind choosing the dual-porosity model for coarse sand and gravel layers? At the first sight, it’s surprising as all three used in the experiment soil materials do not have two-tier structure justifying using the dual-porosity model.

Table 1 Authors described how saturated conductivities of soil materials were determined. But how the parameters of SWRC of these materials were determined?

Reviewer 3 Report

This article describes simulations of drainage in a soil containing artificial capillary barriers, which are important potential solutions to improve agricultural conditions. In order to validate their model, the authors compare their numerical results to experimental results. My main concern is the very small amount of experimental data. Indeed, the authors present results from one reference soil without barrier and two soils with a barrier (actually three but one failed - which is not relevant to indicate unless you manage to get the data).

I will give some general comments about the rest of the paper but I do not think that the paper can be accepted with such few experimental data.

The introduction is interesting until the advertisement of HYDRUS is made in many details which are not needed for introducing this study. An interesting angle would be to mention the other types of transport model that exist and comment on their differences.The objectives of the study are not very clear, please clarify (the last paragraph before materials and results).

The material and methods section is very brief on the experimental data, and it is unclear to me if this data has already been published in another article. More details about the type of soil and the grain size of the sand and gravel are necessary.

I won't comment on the model details since it is not my area of expertise. I didn't understand the choice of all parameters and I am not sure if it is very well explained, although since it is not my area I won't say more on this.

It was hard to follow the description of all simulations that are presented. A table summarizing would be appreciated and would help the reader follow all the different abbreviations.

In the results and discussion part, it is hard to follow because of the before mentioned issues. In Figure 3, it seems obvious that the barriers do not impact the water transport as much as the model describes it. It is also surprising that the model doesn't describe so well the water transport of the soil without barrier. Maybe the initial soil parameters should be adjusted?

The choice of scenarios for the simulations is interesting and meaningful. However the description of the results is hard to follow, maybe because of all the abbreviations. Improving this would help understanding of your analysis.

Round 2

Reviewer 3 Report

The authors have made many adequate changes. In particular, I find the methods section much more detailed and clearer. I would recommend providing more experimental data but the focus is clearly on the simulations here.

I noticed a typo: line 188, "paly" instead of "play".